# The effects of introgression across thousands of quantitative traits revealed by gene expression in wild tomatoes

Mark S. Hibbins[1]*, Matthew W. Hahn[1,2]

1 Department of Biology, Indiana University, Bloomington, Indiana, United States of America, 2 Department of Computer Science, Indiana University, Bloomington, Indiana, United States of America

* mhibbins@iu.edu

**Data Availability Statement:** Raw sequencing reads for the gene expression dataset are available on the SRA under BioProject PRJNA714065. Quantitative expression data, as well as scripts for

## Abstract

It is now understood that introgression can serve as powerful evolutionary force, providing genetic variation that can shape the course of trait evolution. Introgression also induces a shared evolutionary history that is not captured by the species phylogeny, potentially complicating evolutionary analyses that use a species tree. Such analyses are often carried out on gene expression data across species, where the measurement of thousands of trait values allows for powerful inferences while controlling for shared phylogeny. Here, we present a Brownian motion model for quantitative trait evolution under the multispecies network coalescent framework, demonstrating that introgression can generate apparently convergent patterns of evolution when averaged across thousands of quantitative traits. We test our theoretical predictions using whole-transcriptome expression data from ovules in the wild tomato genus *Solanum*. Examining two sub-clades that both have evidence for post-speciation introgression, but that differ substantially in its magnitude, we find patterns of evolution that are consistent with histories of introgression in both the sign and magnitude of ovule gene expression. Additionally, in the sub-clade with a higher rate of introgression, we observe a correlation between local gene tree topology and expression similarity, implicating a role for introgressed *cis*-regulatory variation in generating these broad-scale patterns. Our results reveal a general role for introgression in shaping patterns of variation across many thousands of quantitative traits, and provide a framework for testing for these effects using simple model-informed predictions.

## Author summary

It is now known from studying large genetic datasets that species often hybridize and cross with each other over many generations – a phenomenon known as introgression. Introgression introduces new genetic variation into a population, and this variation can cause traits to be shared among the introgressing species. When researchers study the evolution of trait variation among species, this source of trait sharing is rarely accounted for. Here, we present a statistical model of the effects of introgression on trait variation. This model predicts that, when averaged across many thousands of traits, introgressing species

all analyses, are available from https://github.com/
mhibbins/intro_quant_traits.

**Funding:** This work was supported by a grant
(DEB-1936187) from the National Science
Foundation (https://www.nsf.gov/) awarded to M.
W.H. The funders had no role in study design, data
collection and analysis, decision to publish, or
preparation of the manuscript.

**Competing interests:** The authors have declared
that no competing interests exist.

are consistently more similar than expected from standard approaches. Researchers study-
ing gene expression often consider the expression of many thousands of genes, making
this a case where the expected effects of introgression are likely to manifest. We tested our
model prediction using ovule gene expression data from the wild tomato genus *Solanum*,
in two groups of species with evidence of historical introgression. We found that patterns
of expression similarity in both groups are consistent with their histories of introgression
and the predictions from our model. Our results highlight the importance of accounting
for introgression as a source of trait variation among species.

## Introduction

Introgression—the historical hybridization and subsequent backcrossing of previously isolated
lineages—has come to the forefront of phylogenomics with the availability of genome sequenc-
ing (reviewed in [1,2]). Introgression has been recognized as a powerful and frequent source of
adaptive variation, with many charismatic examples including wing pattern mimicry in butter-
flies [3,4], coat color in snowshoe hares [5], herbivore resistance in sunflowers [6], high-alti-
tude adaptation in humans [7], and fruit color in wild tomatoes [8]. Introgressed alleles do not
have to underlie discrete traits to influence the course of evolution: alleles that contribute to
quantitative trait variation can also lead to more similarity than expected between the intro-
gressing lineages [9].

Empirically investigating the effects of introgressed ancestry on quantitative trait evolution
remains a challenge, despite recent theoretical and methodological advances [9–11]. This is
because many processes besides introgression can shape the distribution of any particular char-
acter, including incomplete lineage sorting and convergence. It is therefore necessary to sample
a large number of traits in order to demonstrate a genome-wide effect of introgression. Gene
expression is commonly used in comparative analyses between species [12–14], allowing for the
study of thousands of quantitative traits in a phylogenetic framework. Introgressed variants act-
ing on gene expression either in *cis* or in *trans* may affect the evolution of gene expression across
the genome. This could have potentially deleterious effects on fitness, which would be consistent
with previous evidence for widespread selection against introgressed alleles [15–18].

Incomplete lineage sorting (ILS) and introgression both introduce shared history that could
influence the evolution of quantitative traits, though neither of these processes are captured by
a standard species phylogeny. Therefore, to paint a complete picture of trait variation among
species, it is necessary to include these sources of topological discordance in order to avoid
errors inherent to methods that typically only consider the species topology [19]. Mendes et al.
[20] showed that when gene tree discordance is unaccounted for, standard comparative
approaches will return inflated evolutionary rate estimates and will underestimate phyloge-
netic signal. Despite these challenges, no approach has included all sources of gene tree discor-
dance into a single framework for quantitative trait evolution. Some methods have extended
the classic Brownian motion model for quantitative trait evolution to include shared histories
due to ILS alone [20], while other work has applied the Brownian motion model to a phyloge-
netic network with introgression but no ILS [9]. A method including both sources of discor-
dance would provide a complete picture of the most common causes of shared evolutionary
history and their effects on quantitative traits. This would in turn allow for more accurate
inferences of key evolutionary parameters, such as the trait evolutionary rate.

To address the effects of historical introgression on quantitative traits, we first develop a
Brownian motion model of trait evolution that includes both ILS and introgression, showing
the expected effects of introgression on the similarity in quantitative traits across species. This

model leads directly to predictions about patterns of trait-sharing on a three-taxon tree, which we test by leveraging whole-transcriptome gene expression data [21] from the wild tomato clade in the genus *Solanum*. This clade includes 13 species that have radiated within the last 2.5 million years, and contains high rates of gene tree discordance due to both ILS and introgression [22,23]. Using ovule expression data from two independent species triplets with different levels of introgression, we find that transcriptome-wide patterns of variation in both triplets are consistent with histories of introgression, with quantitatively stronger signals in the sub-clade with greater introgression. Our analyses demonstrate that introgression can have measurable effects across the genome, on thousands of quantitative traits.

## Results

### Brownian motion on a species tree

To accurately model trait variation among species, we require an understanding of the evolutionary history that relates those species, and a model for how traits are expected to evolve given that history. We present results using Brownian motion, a statistical model that is commonly applied to quantitative traits. The evolutionary history relating species has classically been provided by a species phylogeny. Under Brownian motion, the character states on the tips of this phylogeny follow a multivariate normal distribution, with the variance and covariances of this distribution provided by the branch lengths of the phylogeny [24].

Consider a phylogeny of three species with the topology ((A,B),C) (Fig 1). In units of $2N$ generations, species A and B split at time $t_1$, and C split from the ancestor of A and B at time

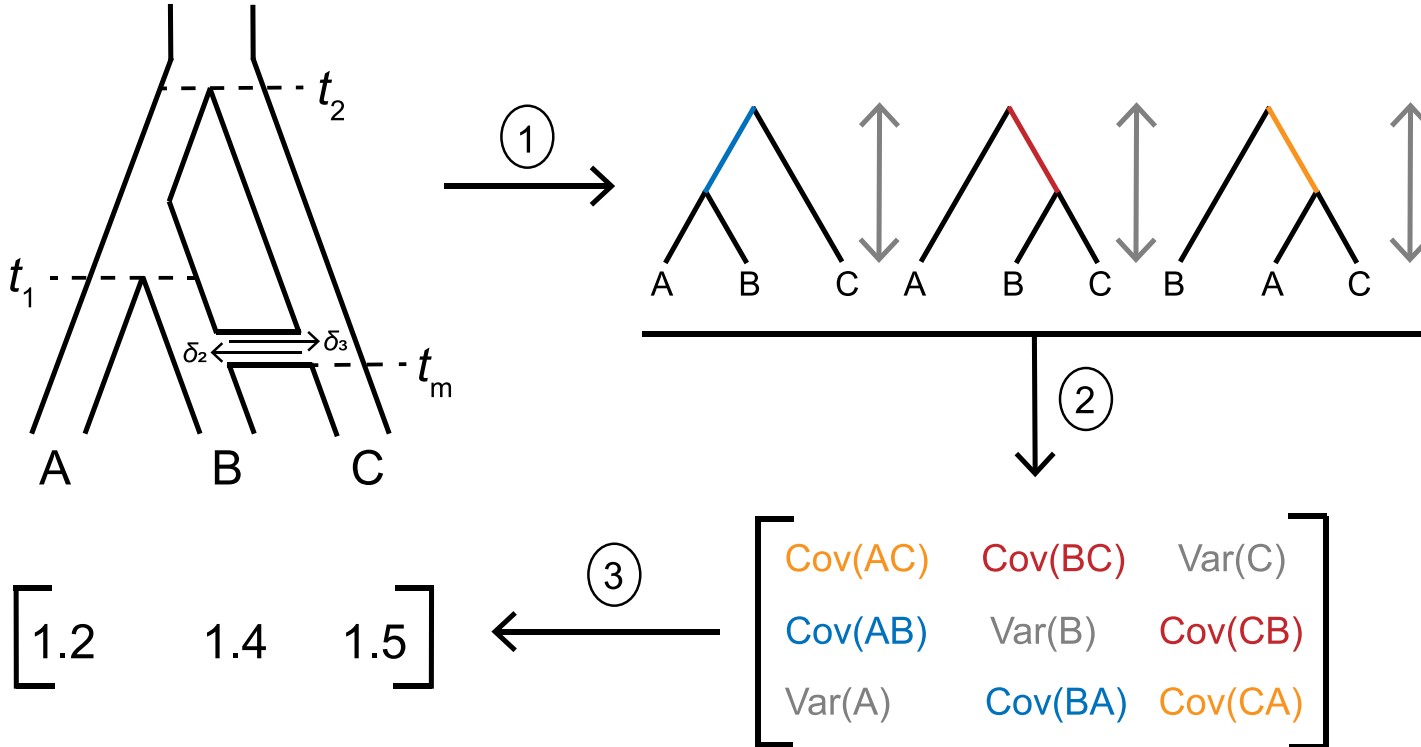

**Fig 1. Modelling quantitative trait evolution under the combined effects of ILS and introgression.** 1) From a phylogenetic network with known parameters, the multispecies network coalescent model can be used to predict the expected frequency and branch lengths of each gene tree topology. 2) These gene trees contribute to trait covariances through their internal branches, and to trait variances through their total heights. The contribution of each gene tree to the overall quantities in $T$ is weighted by its expected frequency. 3) Once the values of $T$ are estimated, character states under Brownian motion can be simulated by drawing from a multivariate normal with a mean of 0 and variance of $\sigma^2 T$.

$t_2$. The expected phylogenetic variances and covariances for three species can be expressed in a 3x3 matrix, which we denote $T$:

$$T = \begin{bmatrix} Var(A) & Cov(AB) & Cov(AC) \\ Cov(BA) & Var(B) & Cov(BC) \\ Cov(CA) & Cov(CB) & Var(C) \end{bmatrix} \quad (1)$$

This matrix is multiplied by the evolutionary rate parameter of the Brownian motion model, $\sigma^2$, to obtain trait variances and covariances. When only the species phylogeny is considered (i.e. there is no ILS or introgression), the trait variances (the diagonal elements of $T$) are determined by the total time along which evolution can occur for each lineage, so $Var(A) = Var(B) = Var(C) = t_2$. The covariances are determined by the length of shared internal branches. In the species tree, only species A and B share an internal branch, so:

$$Cov(AB|no\ ILS\ or\ introgression) = \sigma^2(t_2 - t_1) \quad (2)$$

Where $\sigma^2$ corresponds to the trait evolutionary rate per unit time, and $Cov(AB) = Cov(BA)$.

In the absence of other processes, the species pairs BC and AC have zero covariance. However, trees inferred at individual loci can disagree with the species phylogeny, in which case these species pairs could have shared internal branches, and therefore non-zero covariances. This widespread phenomenon is known as gene tree discordance [22,25–31] and has multiple biological causes [32]. Gene trees with the topologies ((B,C),A) or ((A,C),B) contain internal branches shared by species B+C and A+C, respectively (Fig 1). This results in non-zero covariance terms between these two species pairs in $T$, covariance that cannot arise from evolution solely on the species phylogeny. The consequence of this discordance is that some traits may be closer in value between species that are not closely related in the species tree. We must therefore include discordance in our model to appropriately capture this trait covariance.

## Modelling the effects of only incomplete lineage sorting on quantitative trait variances and covariances

One of the most common causes of gene tree discordance is incomplete lineage sorting, which occurs when ancestral lineages persist through successive speciation events [33,34]. For a rooted triplet, there are four possible gene trees in the presence of ILS: one concordant tree that occurs by lineage sorting with probability $1 - e^{-(t_2 - t_1)}$, and three trees produced by ILS, each with probability $\frac{1}{3}e^{-(t_2 - t_1)}$. One of the three ILS trees is concordant, while the other two are discordant. These probabilities are the basis for the multispecies coalescent model. To obtain the expected trait variances and covariances in $T$, Mendes et al. [20] weight the expected gene tree heights and internal branch lengths, respectively, by their expected frequencies under the multispecies coalescent model. We present those results here with a slightly different formulation for consistency with the new results presented below. For the covariance between A and B, we have:

$$Cov(AB|no\ introgression) = \sigma^2\left[\left(1 - e^{-(t_2-t_1)}\right)\left(\frac{e^{t_2}(t_2 - t_1)}{e^{t_2} - e^{t_1}}\right) + \left(\frac{1}{3}e^{-(t_2-t_1)}\right)\right] \quad (3)$$

In Eq 3, $\sigma^2$ corresponds to the trait evolutionary rate per $2N$ generations, which is the scale over which time is measured in the multispecies coalescent model. Inside the square brackets, the first term is the probability of the gene tree produced by lineage sorting, multiplied by that tree's expected internal branch length in units of $2N$ generations. The second term is the

probability of the concordant tree produced by ILS, which has an expected internal branch length of 1 in units of 2$N$ generations.

Species pairs BC and AC can only have covariance from discordant trees produced by ILS, which gives:

$$Cov(BC|no\ introgression) = Cov(AC|no\ introgression) = \sigma^2\left(\frac{1}{3}e^{-(t_2-t_1)}\right) \tag{4}$$

Again, in these trees the internal branches are of length 1 in units of 2$N$ generations, and so are not shown explicitly.

For the expected trait variances, all three species share the same expected variance, which is the total height of all the gene trees weighted by their probabilities. These are:

$$Var(A|no\ introgression) = Var(B|no\ introgression) = Var(C|no\ introgression)$$
$$= \sigma^2[(1 - e^{-(t_2-t_1)})(t_2 + 1) + (e^{-(t_2-t_1)})(t_2 + 1 + 1/3)] \tag{5}$$

Where the first term in the square brackets is the contribution from the lineage sorting tree, and the second term is the contribution from the three ILS trees.

## Modelling the effects of introgression and ILS on quantitative trait variances and covariances

Now, we extend these expressions for species variances and covariances to include both ILS and introgression. We envision an instantaneous introgression event between species B and C (Fig 1), which occurs at time $t_m$. This event can be in either direction, with the probabilities of a locus following a history of C → B introgression or B → C introgression represented using $\delta_2$ or $\delta_3$, respectively. To capture the processes of ILS and introgression simultaneously, we imagine that each possible history at an individual locus can be represented by a "parent tree" within which lineage sorting or ILS occurs according to the multispecies coalescent process [10,35–37]. This is sometimes referred to as the multispecies network coalescent [38,39]. For our model, we consider three parent trees (see S3 Fig): one with no introgression, which occurs with probability 1 − ($\delta_2 + \delta_3$), and two parent trees for the two possible directions of introgression, which occur with probabilities of either $\delta_2$ or $\delta_3$ (these probabilities represent the "rate" of introgression in our model). Each of these three parent trees can generate four possible gene trees with three possible topologies (Fig 1, arrow 1), which vary in the frequency of topologies and expected branch lengths depending on each parent tree's parameters (as in the model of ILS-only described in the previous section).

To obtain expressions for the expected variances and covariances under this model, we must sum the contributions of all gene trees within each parent tree, and then sum the contribution of each parent tree (Fig 1, arrow 2). For the covariance between A and B, this gives:

$Cov(AB|ILS\ and\ introgression)$

$$= \sigma^2\left[(1 - (\delta_2 + \delta_3))\left[(1 - e^{-(t_2-t_1)})\left(\frac{e^{t_2}(t_2 - t_1)}{e^{t_2} - e^{t_1}}\right) + \left(\frac{1}{3}e^{-(t_2-t_1)}\right)\right] + \delta_2\left(\frac{1}{3}e^{-(t_2-t_m)}\right) + \delta_3\left(\frac{1}{3}e^{-(t_1-t_m)}\right)\right] \tag{6}$$

Note that the term inside the inner square brackets in Eq 6 is the same as in Eq 3, but is now weighted by the probability of a history with no introgression. In addition, there are two additional terms denoting the contributions of trees generated by ILS that follow a history of introgression (because ILS occurs regardless of the history at a locus). For a complete derivation, including the expectations of each gene tree within each parent tree, see S1 Text.

For the covariance between B and C, we have:

$$Cov(BC|\text{ILS and introgression})$$
$$= \sigma^2\left[(1-(\delta_2+\delta_3))\left[\frac{1}{3}e^{-(t_2-t_1)}\right]+\delta_2\left[(1-e^{-(t_2-t_m)})\left(\frac{e^{t_2}(t_2-t_m)}{e^{t_2}-e^{t_m}}\right)+\left(\frac{1}{3}e^{-(t_2-t_m)}\right)\right]+\delta_3\left[(1-e^{-(t_1-t_m)})\left(\frac{e^{t_1}(t_1-t_m)}{e^{t_1}-e^{t_m}}\right)+\left(\frac{1}{3}e^{-(t_1-t_m)}\right)\right]\right] \quad (7)$$

Introgression occurs between B and C in our model, so B and C are sister in the parent trees that represent the two directions of introgression (see S1 Text, S3 Fig). This means that these parent trees can each produce two gene trees with BC as sister species: one from lineage sorting and one from ILS. The contributions of these two gene trees in each parent tree are captured in the last two terms of Eq 7. The first term corresponds to the contribution of ILS from the parent tree without introgression, i.e. Eq 4.

Finally, for the covariance between A and C, we have

$$Cov(AC|\text{ILS and introgression}) = \sigma^2\left[(1-(\delta_2+\delta_3))\left(\frac{1}{3}e^{-(t_2-t_1)}\right)+\delta_2\left(\frac{1}{3}e^{-(t_2-t_m)}\right)+\delta_3\left(\frac{1}{3}e^{-(t_1-t_m)}\right)\right] \quad (8)$$

Since gene trees where A and C are sister can only be produced by ILS in our model, Eq 8 is simply the sum of the gene trees with this topology produced by each of the three parent trees.

Lastly, we consider the expected trait variance with introgression. As with the covariances, we sum the total contribution of each gene tree within a parent tree, and then sum these contributions across each parent tree. All three share the same gene tree heights and therefore have the same expected variances. This gives:

$$Var(A) = Var(B) = Var(C)$$
$$= \sigma^2[(1-(\delta_2+\delta_3))[(1-e^{-(t_2-t_1)})(t_2+1)+(e^{-(t_2-t_1)})(t_2+1+1/3)]+\delta_2[(1$$
$$-e^{-(t_2-t_m)})(t_2+1)+(e^{-(t_2-t_m)})(t_2+1+1/3)]+\delta_3[(1-e^{-(t_1-t_m)})(t_1+1)$$
$$+(e^{-(t_1-t_m)})(t_1+1+1/3)]] \quad (9)$$

The first term represents the contribution of the parent tree with no introgression, the same as in Eq 5. The second two terms represent the contributions to the total variance from $C \rightarrow B$ and $B \rightarrow C$ introgression, respectively. When $T$ is updated to include all these expectations, it becomes possible to model character states under Brownian motion while accounting for both ILS and introgression.

## Testing for the effect of introgression on quantitative traits

To evaluate whether patterns of quantitative trait variation are consistent with a history of introgression, we use a simple test statistic that employs the same logic as the $D_3$ test for introgression [40]; see also the $f_3$ statistic [41]. Imagine that species A, B, and C have values $q_1$, $q_2$, and $q_3$ for a hypothetical quantitative trait, respectively. Given the species tree ((A,B),C), and assuming the Brownian motion model of trait evolution described in the previous sections, the expected distance between trait values $q_2$ and $q_3$ should be equal to the expected distance between $q_1$ and $q_3$. This is because species C is equidistant to species A and B in the phylogeny, and this tree determines quantitative trait variances and covariances. The same relationship between distances is expected when considering the ILS-only model, because of symmetries in expected gene tree frequencies and branch lengths, and therefore in trait covariances (see Eq 4).

However, introgression can introduce additional covariance between one pair of species, resulting in that pair having more similar trait values than the other non-sister pair (see Eqs 7

and 8). This naturally leads to the following test statistic:

$$Q_3 = \frac{|q_2 - q_3| - |q_1 - q_3|}{|q_2 - q_3| + |q_1 - q_3|} \tag{10}$$

The numerator of $Q_3$ takes the difference in trait distances between the two pairs of non-sister species; when there is no introgression, this numerator—and therefore $Q_3$—has an expected value of 0. When a significant non-zero value of $Q_3$ is observed, the statistic is consistent with a history of introgression. In addition, the sign of the statistic can tell us which species were involved in introgression (but not the direction of introgression). For example, a negative $Q_3$ value would be consistent with introgression between species B and C, since that would result in $q_2$ and $q_3$ having more similar values (and therefore a smaller distance between them). The denominator of $Q_3$ is the sum of the two trait distances, which normalizes the statistic between 0 and 1, allowing it to be compared across traits with different mean values. We imagine that this statistic will be applied to many individual quantitative traits, each providing a separate value of $Q_3$. The significance for a dataset consisting of many traits can then be evaluated either by testing for a mean value of $Q_3$ significantly different from 0, or by using a sign test with the null expectation that positive and negative $Q_3$ values should be equally frequent (see the analyses below for more details).

To confirm the effects of introgression predicted by the model, and the ability of $Q_3$ to detect it, we performed a power analysis. First, to illustrate the conceptual basis for $Q_3$, we contrasted two conditions: an ILS-only condition and an ILS + introgression condition (Fig 2). Both scenarios use the three-taxon tree described in previous sections, simulating quantitative traits as the sum of contributions of many genes (and therefore gene trees; see Materials and Methods). For 20,000 independent simulated traits we calculated the mean and standard error of the difference in trait value at the tips of the tree between each pair of species (Fig 2). As predicted by our model, the taxa involved in introgression had a higher covariance and more similar trait values than the non-introgressing pair of taxa when averaging across the 20,000 traits (Fig 2).

Second, we performed a power analysis across 90 different parameter combinations: three values each of the timing of introgression, the level of ILS, and the number of genes, and four values of the rate of introgression. We simulated 100 datasets for each set of parameters and asked how often $Q_3$ was significantly different from 0 in the direction predicted by introgression. We found the most important parameter to be the rate of introgression: at a rate of 1% (i.e. 1% of the genome has been introgressed), power was consistently low (1-6%) regardless of other simulation parameters (Figs 3 and S1). At higher rates of introgression, power was increased when introgression was more recent relative to speciation, when the level of ILS was lower, and when more genes (traits) were considered. When 5,000 genes were used, power reached 67% under the best-case scenario (S1 Fig); this increased to 97% with 15,000 genes (Fig 3). Simulations under a no-introgression scenario yielded false positive rates of less than 5% across all conditions (S2 Fig).

## Gene expression variation is consistent with inferred histories of introgression in *Solanum*

We used previously generated introgression and gene expression datasets from the wild tomato clade, *Solanum* section *Lycopersicon*, to empirically evaluate the effects of introgression on thousands of expression traits. This clade includes the domesticated tomato, *S. lycopersicum*, and its 12 wild relatives, which have all originated in the last 2.5 million years. The first dataset is a phylogenetic analysis of 29 accessions (i.e. populations) across these 13 tomato species and two outgroups [22]. This dataset includes an introgression analysis based on *D*-

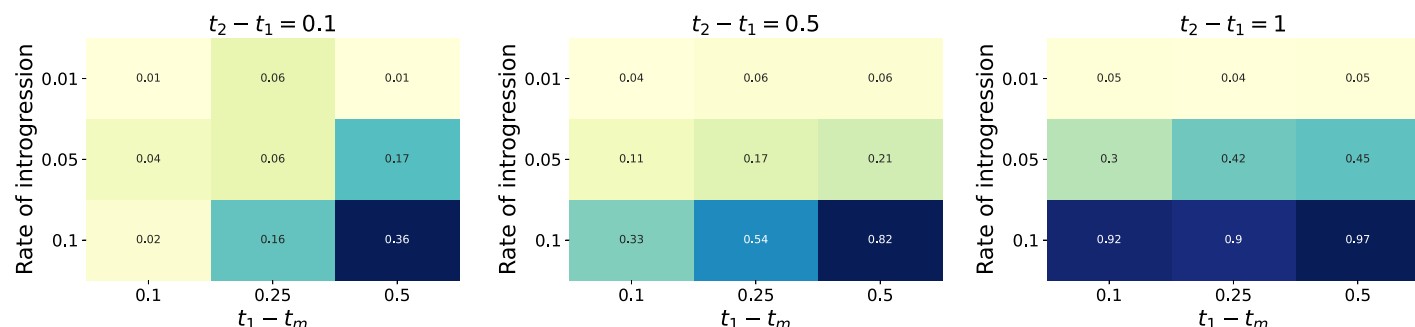

**Fig 2. Quantifying the effect of introgression on quantitative trait variation.** For ILS-only (top row) and ILS + introgression (bottom row) conditions, we show the expected variance/covariance matrix (middle-left column, variances not shown for clarity) and the average difference in quantitative trait values between each pair of species across 20,000 simulated traits (middle-right column). The expectations for the $Q_3$ statistic are also shown (far-right column).

**Fig 3. Power analysis of the ability of $Q_3$ to detect a signature of introgression from 15,000 simulated genes ($\sigma^2 = 1$).** Each cell reports the proportion of 100 simulated datasets where $Q_3$ was significantly different from 0 in the direction expected from the simulated history of introgression. Within each matrix, the x-axis is the time of introgression relative to speciation (larger values mean relatively more recent introgression), and the y-axis is the rate of introgression. There is one matrix for each of three times between speciation events, which determine the levels of ILS (decreasing from left to right, as the times increase). The greatest power comes in scenarios with little ILS, high rates of introgression, and recent introgression events.

statistics [42,43] across all possible quartets, which provides a comprehensive overview of patterns of introgression in the clade. The second dataset is normalized quantitative expression of 14,556 genes expressed in ovules from six accessions across five tomato species. This includes published data for five accessions across four species [21], while data from the other two species are previously unpublished. Expression levels for each gene are represented as reads per kilobase of transcript, per million mapped reads (RPKM). Samples were collected on the day of flower opening for 1-4 individuals of each species grown in a common greenhouse [21].

Combining these two datasets, we sought to identify triplets of species with both evidence of introgression (from sequence data) and available gene expression data, so that we could apply the $Q_3$ statistic. Additionally, we wanted these triplets to vary in the magnitude of introgression, so that the magnitude of the effect of introgression on trait variation could be evaluated in addition to the presence or absence of an effect. With these considerations in mind, we identified two triplets. The first consists of the accessions *LA3475* (*S. lycopersicum*), *LA1589* (*S. pimpinellifolium*), and *LA0716 (S. pennellii)*, with *LA3475* and *LA1589* as sister taxa, and evidence of introgression between *LA1589* and *LA0716* ($D = 0.057$, $P = 0.0015$; values from [22]) ([Fig 4A]). Using the $D_p$ statistic [23] on site pattern counts from [22], we obtained a value of 0.0013, corresponding to a genomic rate of introgression of 0.13%. We hereafter refer to this triplet as the "low" triplet because of the relatively low observed rate of introgression. The other triplet consists of *LA3778* (*S. pennellii*), *LA1777* (*S. habrochaites*), and *LA1316* (*S. chmielewskii*), with *LA3778* and *LA1777* as sister taxa, and significant introgression between *LA1777* and *LA1316* ($D = 0.135$, $P = 2.34 \times 10^{-35}$; values from [22]) ([Fig 4A]). We obtained a $D_p$ value of 0.0744 for this triplet, corresponding to a rate of introgression of 7.44%; this value is likely an underestimate, as $D_p$ tends to underestimate the true value at higher rates of introgression [23]. As the rate of introgression is much higher for this triplet, we refer to it as the "high" triplet.

We used expression values from 14,556 genes available in both the low and high triplets. For each gene we calculated a separate value $Q_3$, averaging across genes to obtain a mean value for each triplet. We obtained transcriptome-wide mean $Q_3$ values of -0.012 and -0.019 for the low and high triplet, respectively ([Fig 4B]). The values we observe are consistent with the histories of introgression inferred from the sequence data in both sign and magnitude. Both triplets have negative values, which is consistent with introgression between *S. pimpinellifolium* and *S. pennellii* in the low triplet, and between *S. habrochaites* and *S. chmielewskii* in the high triplet (see [Fig 4A] for the accessions assigned as $q_1$, $q_2$, and $q_3$ in each triplet). The $Q_3$ value is also more negative in the high triplet, which is consistent with the higher level of introgression inferred from sequence data.

The signal of introgression from quantitative traits was also statistically significant in both triplets, using either method for assessing significance. Using a bootstrapping approach to ask whether the mean values were different from 0 (see Materials and Methods), we obtained $P = 0.0012$ and $P < 0.0001$ for the low and high triplets, respectively ([Fig 4B]). We obtained similar results when testing for a significant excess of either positive or negative $Q_3$ values (i.e. a sign test) at individual genes using bootstrapping ([Fig 4C]; see Materials and Methods). For the low triplet, we observed 7432 negative and 7124 positive genes ($P = 0.0134$); for the high triplet, 7533 negative and 7020 positive genes ($P < 0.0001$). Again, the larger number of negative $Q_3$ values in the high triplet is consistent with a higher amount of introgression.

## Gene-level analysis of expression data

The expression level of genes can be affected by either *cis*-acting or *trans*-acting variants. Because *cis*-acting variants are most often found near the gene they affect [44,45], we might

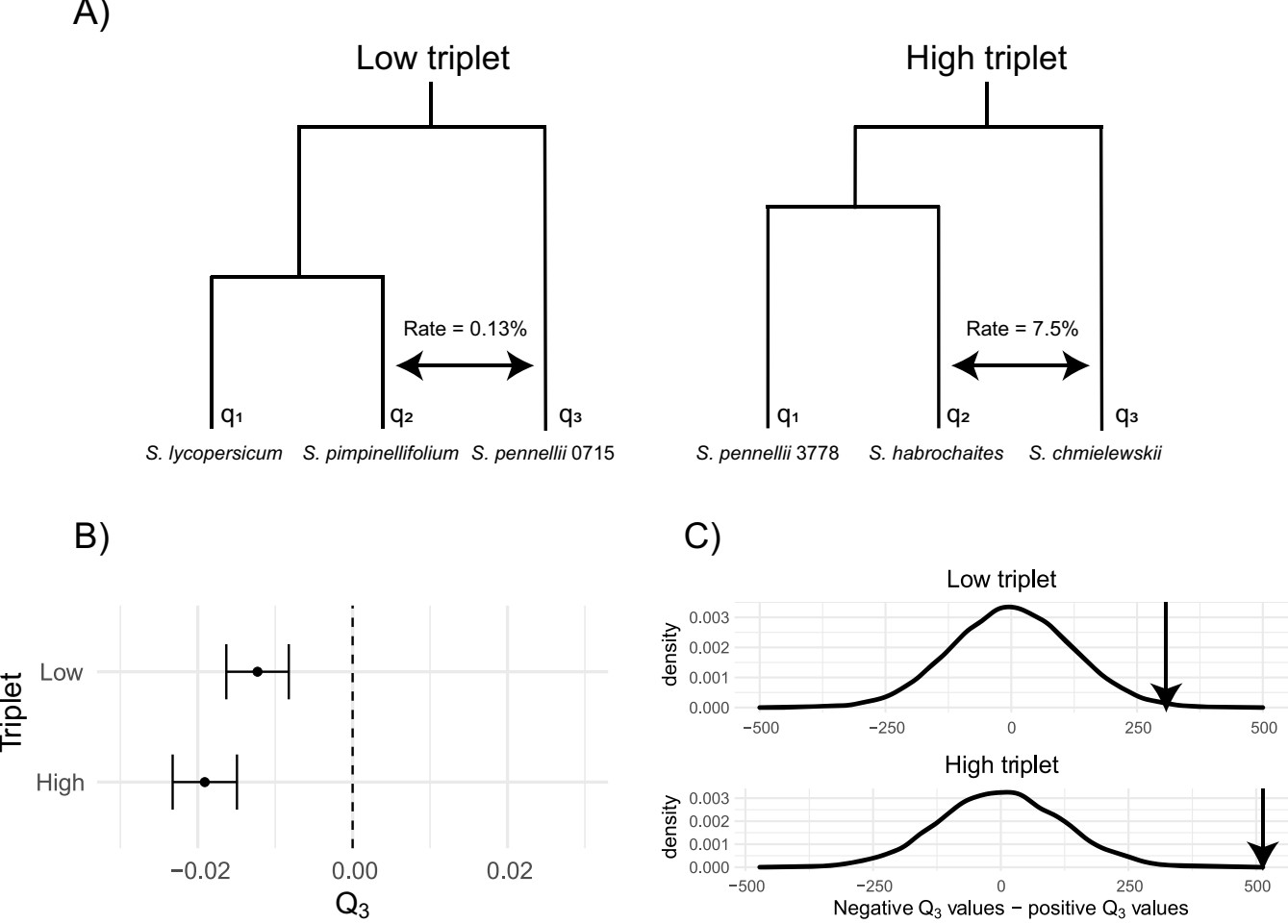

**Fig 4. Ovule gene expression variation in tomatoes is consistent with inferred histories of introgression.** A) Histories of speciation and introgression for our chosen triplets in *Solanum*. B) Mean and standard error of $Q_3$ across all genes in each triplet. C) Difference in the number of genes with a negative vs. positive $Q_3$ value for both triplets. Density plots show the distribution of this difference across 10,000 bootstrapped datasets. Observed values for the two triplets relative to the bootstrap distributions are shown with arrows.

expect these regulatory elements to share the same local gene tree topology as the nearby genic protein-coding region; any signature of introgression would likely be reflected in both regions. While recombination either before or after introgression will uncouple the tree topology in the regulatory region from that in the coding region, we might expect to see an association between patterns of similarity in expression levels and patterns of gene tree discordance if *cis*-acting variants are common.

To test for such a relationship, we looked for an association between coding-region tree topologies and expression similarity among species in both triplets. Using trees estimated from each protein-coding gene [22], we identified 11,061 genes for which both the tree topology and expression values from all species were available. For each gene, we obtained the rooted tree topology for the relevant triplet and also determined which pair of species was most similar in expression value. We assume that expression similarity reflects the local topology at whichever locus has the largest effect on expression, such that the most similar pair of species represents the sister species in this topology.

In the low triplet, we found no significant relationship ($P = 0.776$, $\chi^2$ test of independence) between protein-coding gene tree topology and expression similarity (Fig 5A). In the high triplet, however, we did observe a significant relationship ($P = 0.019$, Fig 5B). For gene trees with a topology consistent with introgression (where *S. habrochaites* and *S. chmielewskii* are sister), there were significantly more genes where expression was also most similar between these species than expected by chance (476 observed vs. 449 expected). In other words, we found that gene expression similarity is correlated with the tree topology of protein-coding genes in the high triplet, in a fashion consistent with *cis*-acting effects of introgressed variation on expression.

## Discussion

Phylogenetic comparative methods provide powerful tools for studying the origins of trait variation among species. However, the rampant gene tree discordance uncovered in many phylogenomic studies paints a more complicated picture of the shared history among species. To date, most models of trait evolution employed by comparative methods have assumed that only the species phylogeny contributes to trait covariance, and have ignored covariance due to discordant gene trees. Our model builds on previous work [9,20] to incorporate both ILS and introgression into a single framework that captures the most common causes of discordance and their effects on quantitative trait evolution. We show that introgression leads to more discordance and stronger patterns of covariance in quantitative traits among non-sister species than ILS alone, paralleling results for binary traits under the same multispecies network coalescent framework [10].

Our model makes several assumptions and simplifications related to expected levels of genetic covariance between species. First, we have modeled post-speciation introgression as a single instantaneous pulse of exchange between one pair of non-sister species. Many other possible introgression scenarios are possible, such as multiple pulses or continuous periods of gene flow. Although each of these scenarios will increase the variance in gene tree topologies, we expect that they will still leave a detectable signature on quantitative traits because they still lead to gene tree asymmetries. In contrast, other gene flow scenarios—such as introgression between sister taxa, or between both pairs of non-sister taxa in a triplet at equal rates—will not result in a detectable signature of gene tree asymmetry. Second, we have assumed that the expected frequencies and coalescence times of loci contributing to trait variation follow neutral expectations. Through a local reduction in $N_e$, directional selection may reduce the rate of gene tree discordance due to ILS, while increasing the rate of discordance due to introgression [18,46,47]. This increase in the rate of introgression relative to ILS may allow for greater power to detect a signal of introgression in quantitative traits, as we show in our power analysis. This implies that positive selection, especially on introgressed variants [48], will make it more likely for quantitative traits to covary between non-sister taxa.

We also make key assumptions about the model of trait evolution. Our model assumes that traits evolve under a Brownian motion process, rather than alternative processes such as the Ornstein-Uhlenbeck (OU) [49] or early-burst [50,51] models. While it may be uncommon for traits to evolve according to an early-burst model [52], many quantitative characters are likely to be constrained in some way, which can be modelled by the OU process. For gene expression in particular, evidence suggests that over long phylogenetic timescales the OU process is a better fit to the data [53–55]. However, multiple non-biological factors may favor the fit of OU models over Brownian motion, including small amounts of error in measured quantitative traits [56]. While we do not expect the model of trait evolution to affect asymmetries between species in thousands of traits, future work incorporating additional models of trait evolution, and their effect on trait covariances in particular, would be useful.

# A) Low triplet

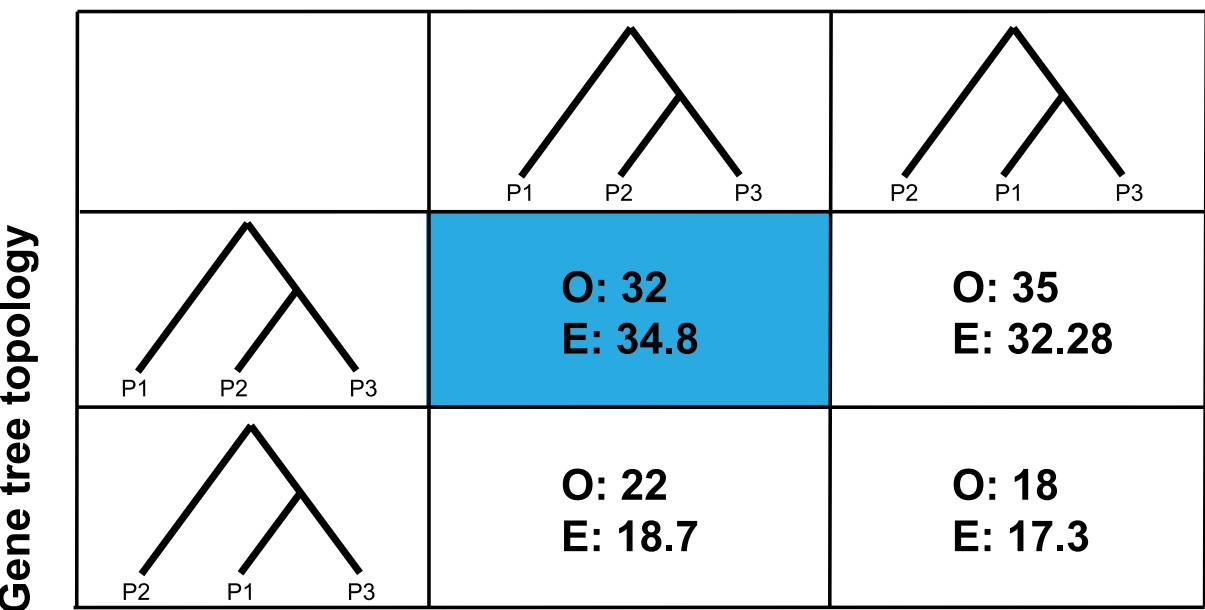

# B) High triplet

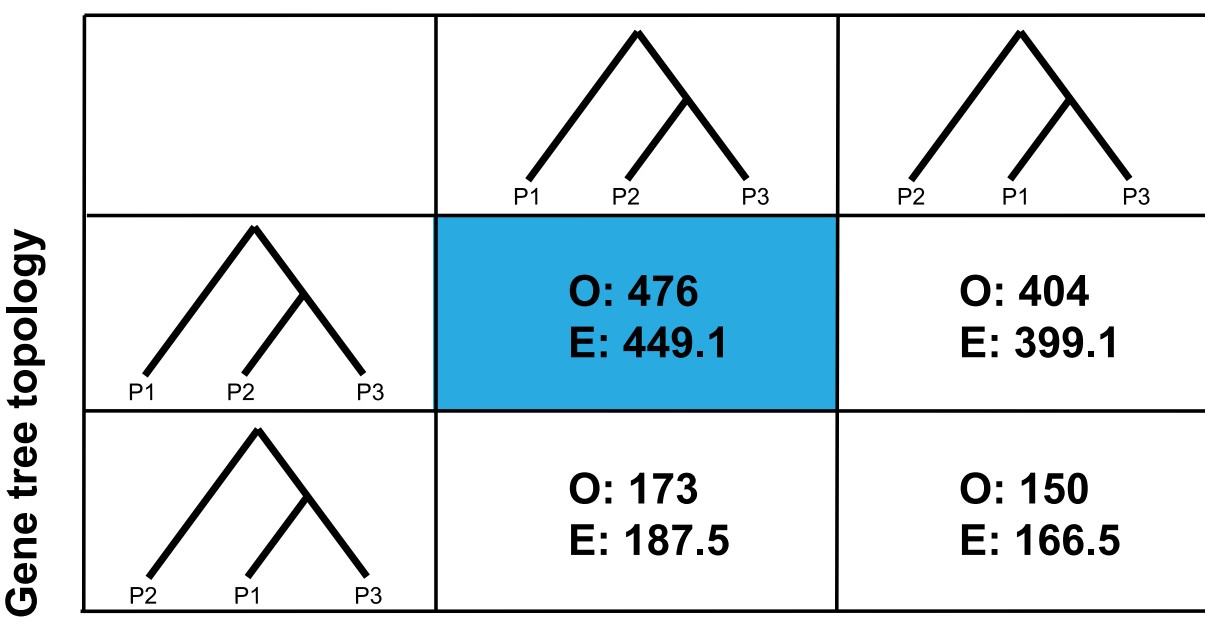

**Fig 5.** Relationships between coding sequence tree topology (rows) and gene expression similarity (columns) in the low (A) and high (B) triplets. Note that for expression similarity, we did not explicitly construct trees from expression data—the tree representation is simply meant to depict observed expression distances. Only discordant trees and expression patterns are shown, but $\chi^2$ P-values (0.776 and 0.019 for panels A and B, respectively) are reported from the full 3x3 table (see S1 and S2 Tables for the full tables). The cases where both the tree topology and pattern of expression are consistent with the inferred history of introgression for that triplet are highlighted in blue. Each cell reports the observed number of genes (O) in each category, and the number expected (E) from the $\chi^2$ distribution.

A key assumption of our statistical analysis is that each gene expression trait evolves independently. However, many genes show correlated patterns of expression, either because of locally shared *cis*-acting elements or because of *trans*-acting factors that affect the expression of many genes across the genome [44,45]. If, for instance, such a *trans*-acting factor is introgressed and affects many genes in a similar way, then treating each gene as an independent observation would constitute pseudoreplication of measurements. However, there are two pieces of evidence that suggest pseudoreplication is not a major problem in our analyses. First, previous data from experimental introgression lines between *S. lycopersicum* and *S. pennellii* are not consistent with a large role of introgressed loci on background gene expression: Guerrero et al. (2016) [57] found that each introgressed gene had downstream effects on the expression of only 0.4 genes on average. Second, we find here that the number of genes where expression is more similar between introgressing species is higher in the triplet with a higher rate of introgression. This is again consistent with largely local effects of each introgressed locus on gene expression. Based on these observations, we conclude that we likely have many thousands of independent data points testing the relationship between introgression and expression variation, even if the true correlation structure is unknown.

Our power analysis suggests that we should have had low power to detect an effect of introgression in the low triplet, which has a rate of introgression of less than 1%. One explanation for the fact that we do detect an effect is that the introgressed variation in this triplet affects the downstream regulation of a large set of correlated genes, though the discussion in the previous paragraph likely rules out this possibility. Very recent introgression is also an unlikely explanation, as our power analysis shows that the timing of introgression does not have an effect at low rates. As previously discussed, directional selection on introgressed variation in the low triplet could also improve the power; evaluating this possibility would be an interesting future direction. Finally, we may simply have been fortunate to observe a positive result, even with reduced (but non-zero) power in this area of parameter space. Distinguishing random chance from other processes would also be facilitated by testing additional triplets; unfortunately, we have exhausted the independent triplets possible from our data, having used six of the eight available accessions with gene expression data. Very few multispecies transcriptomic datasets are currently available in systems with widespread introgression, though similar tests may be possible from data in the butterfly genus, *Heliconius* [54,58]. Analyzing or generating such datasets in other systems would help to confirm the generality of our findings.

Our analysis of gene expression is consistent with the idea that introgression between wild tomato species has broadly influenced variation in gene expression among species. An alternative explanation is that species with more similar gene expression may be more likely to introgress, possibly due to reduced negative fitness consequences from hybrid dysregulation. There are again a number of pieces of evidence that argue against the latter interpretation. Guerrero et al. (2016) [57] found no evidence for an association between the magnitude of differential expression between tomato introgression lines and the sterility of hybrids. While those experiments had fewer generations of hybridization than wild introgressed populations—and were conducted in a greenhouse—they do not indicate that general expression levels are a barrier to introgression. Furthermore, here we observe a correlation between expression similarity at

specific genes and the tree topology inferred from their protein-coding sequences (Fig 5B). This association suggests a direct causal effect of introgressed genes on their expression: *cis*-regulatory differences at introgressed loci lead to a relationship between local tree topologies and expression levels (cf. [59]). Such a relationship is highly unlikely to instead be due to a barrier to introgression. The fact that we do not observe the same correlation in the "low" triplet (Fig 5A) could be due either to a comparative lack of statistical power in this triplet, or due to more recombination between the protein-coding regions the tree topologies were inferred from and the *cis*-regulatory regions driving expression. Introgression will reduce the opportunities for recombination, which could explain why the "high" triplet retains a higher signal. Alternatively, it may be that *trans*-acting variation is much more common in this triplet, a scenario that would not lead to an association between local gene tree topologies and local gene expression. We cannot definitively distinguish among these possibilities given only the data presented here. Finally, it is possible that some form of experimental or technical artefact could be responsible for asymmetries in many traits, though we note that the sister species in both triplets examined here always show the greatest similarity in gene expression (S1 and S2 Tables). The association we observe between tree topologies and expression similarity at individual genes is also inconsistent with an artefact.

Overall, our results demonstrate both theoretically and empirically that introgression can affect patterns of quantitative trait evolution. While considerable attention and excitement has justifiably been devoted to the power of introgression as an evolutionary force shaping trait variation, this is a double-edged sword, as most phylogenetic comparative methods do not account for gene tree discordance. Previous work has shown that discordance due solely to ILS can lead to overestimates of the rate of quantitative trait evolution and to underestimates of phylogenetic signal [20]. The effects of introgression in misleading our inferences will be worse, as it both increases overall discordance and generates asymmetries in trait sharing. Future phylogenetic comparative approaches should strive to evaluate the contributions of both ILS and/or introgression on trait evolution, allowing for more accurate evolutionary inferences. Doing so will pave the way for more powerful inferences about the evolutionary forces that shape trait variation among species.

## Materials and methods

### Description of datasets

We use ovule gene expression data that is described in Moyle et al. [21]. The dataset consists of normalized quantitative expression for 14,556 genes measured in six accessions across five species (two different accessions of *S. pennellii* were used in the two triplets). For each accession, samples were collected on the day of flower opening for 1-4 biological replicates (individual plants) grown in a common greenhouse. When applicable, we took the average expression value across replicates within each accession for our analyses. Raw sequencing reads for this dataset are available on the SRA under BioProject PRJNA714065. The dataset containing normalized expression for each replicate, in addition to the scripts for all analyses, are available from https://github.com/mhibbins/intro_quant_traits.

We use phylogenomic data that is described in Pease et al. [22]. The dataset consists of transcriptomes from 29 accessions across 13 species, including the six accessions used in our analyses. Pease et al. [22] used MVFtools to estimate transcriptome-wide *D*-statistics for all possible rooted triplets (2925 total values) across the 27 ingroup accessions. From this dataset we selected the two triplets to use in our analyses. Pease et al. [22] also inferred gene trees for each individual protein-coding region (19,116 genes total) using *RAxML* [60]; we used this data in

our gene-level analyses. Both datasets are published in the Dryad repository https://doi.org/10.5061/dryad.182dv.

## Simulation of quantitative traits & power analyses

We simulated the effect of introgression on quantitative trait values (as shown in Fig 2) under two models: an ILS-only model and a model with ILS and introgression. For the ILS-only model, we used values of 1 and 1.3 for the speciation time of A and B, and the speciation of C from the ancestor of A and B, respectively (all in units of $2N$ generations). The introgression condition maintained the same speciation times, with the addition of an introgression event from C into B at a time of 0.5, with $\delta_2 = 0.1$. Using these parameters, we used our model to construct expected variance/covariance matrices with $\sigma^2 = 1$ using a custom $R$ function (script available at https://github.com/mhibbins/intro_quant_traits/blob/main/scripts/bm_model_sims.R). We then simulated trait values by drawing from a multivariate normal distribution using the $R$ function *mvrnorm* with means of 0 and the constructed matrices.

We performed a power analysis to assess the statistical power of $Q_3$. Using the simulation approach described above, we simulated 100 trait datasets under all combinations of the following parameters: 5000, 10000, and 15000 for the number of genes; 0.1, 0.5, and 1 for $t_2 - t_1$; 0.1, 0.25, and 0.5 for $t_1 - t_m$; and 0, 0.01, 0.05, and 0.1 for the rate of introgression. We evaluated significance for each dataset using a one-sample $t$-test with $H_0$: $Q_3 = 0$. A result was considered a true positive for our analysis when $P < 0.05$ and the sign of the mean simulated $Q_3$ value was consistent with the simulated history of introgression.

## Testing quantitative traits for introgression

We calculated average expression values across individual replicates of each accession before estimating $Q_3$ for each gene. To assess the significance of both our estimated $Q_3$ means and signs, we used bootstrap-resampling. For the mean $Q_3$ values, we tested the null hypothesis of $Q_3 = 0$ by randomly sampling 10,000 datasets of 14,556 genes each with replacement from the empirical gene expression dataset, and estimating the mean value of each. We assessed the rank $i$ of the observed $Q_3$ values among these resampled datasets, and a two-tailed $P$-value was estimated using the following formula:

$$P = 1 - 2 * |0.5 - (i/n)|$$

where $n$ is the number of observations (in this case, 10,000). This formula measures the deviation of the observed value from the center of the bootstrapped dataset, which has a rank of 5000. For the sign of individual genes' $Q_3$ values, we tested the null hypothesis that the number of negative and positive signs are equal by randomly sampling 10,000 datasets of 14,556 genes each. For each resampled dataset we counted the number of negative and positive $Q_3$ values, ranking the datasets from the one with the greatest excess of negative values to the greatest excess of positive values. The rank of the observed data against these resampled datasets was calculated, and two-tailed $P$-values were evaluated using the same formula as above.

For the analysis of the relationship between gene-level tree topology and expression similarity, we made use of gene trees inferred using *RAxML* by ref. [22]. We used the Python package *ete3* [61] to prune these gene trees down to the accessions involved in our test triplets. We then obtained the overlapping set of genes for which both topologies and expression data were available, and recorded the expression "topology" based on the minimum pairwise distance in expression values. The counts of gene tree topology and expression topology were placed into

a 3x3 contingency table for each triplet, and we tested for a significant association using a $\chi^2$ test of independence.

## Supporting information

**S1 Fig.** Power analysis results using (A) 5000 and (B) 10,000 genes.
(EPS)

**S2 Fig. Power analysis results under no introgression (i.e. false positive rates).**
(EPS)

**S3 Fig. Modelling introgression using parent trees.** Parent trees represent the possible histories of speciation and/or introgression at a locus, with their probabilities determined by the rate of introgression. Within each parent tree, gene trees sort according to the multispecies coalescent process. Adapted from [10].
(EPS)

**S1 Table. Full 3x3 contingency table for relationship between gene-level expression and local gene tree topology in the low triplet.**
(DOCX)

**S2 Table. Full 3x3 contingency table for relationship between gene-level expression and local gene tree topology in the high triplet.**
(DOCX)

**S1 Data. Pairwise distances between simulated trait values for 20,000 genes.**
(XLSX)

**S2 Data. P-value and sign of Q3 statistic reported for each combination of parameters in our power analysis, with 15,000 simulated traits.**
(TXT)

**S3 Data. Normalized (RPKM) gene expression data from *Solanum* ovules used in our empirical analyses.**
(CSV)

**S4 Data. Gene tree topology counts for each trio calculated from Pease et al. [22] transcriptomic dataset.** Used in the gene-level expression analysis.
(CSV)

**S5 Data. P-value and sign of Q3 statistic reported for each combination of parameters in our power analysis, under the no-introgression condition.**
(TXT)

**S6 Data. P-value and sign of Q3 statistic reported for each combination of parameters in our power analysis, with 5000 simulated traits.**
(TXT)

**S7 Data. P-value and sign of Q3 statistic reported for each combination of parameters in our power analysis, with 10,000 simulated traits.**
(TXT)

**S1 Text. Derivation of expected trait variances and covariances under the multispecies network coalescent**
(DOCX)

## Acknowledgments

We thank Leonie Moyle and Matthew Gibson for sharing data, and for comments on the manuscript.

## Author Contributions

**Conceptualization:** Mark S. Hibbins, Matthew W. Hahn.

**Data curation:** Mark S. Hibbins.

**Formal analysis:** Mark S. Hibbins.

**Funding acquisition:** Matthew W. Hahn.

**Investigation:** Mark S. Hibbins.

**Methodology:** Mark S. Hibbins.

**Project administration:** Mark S. Hibbins, Matthew W. Hahn.

**Resources:** Mark S. Hibbins, Matthew W. Hahn.

**Software:** Mark S. Hibbins.

**Supervision:** Matthew W. Hahn.

**Validation:** Mark S. Hibbins, Matthew W. Hahn.

**Visualization:** Mark S. Hibbins.

**Writing – original draft:** Mark S. Hibbins, Matthew W. Hahn.

**Writing – review & editing:** Mark S. Hibbins, Matthew W. Hahn.

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
