## [Decision Letter · Decision Letter 0]

19 Sep 2021

Dear Dr Hibbins,

Thank you very much for submitting your Research Article entitled 'The effects of introgression across thousands of quantitative traits revealed by gene expression in wild tomatoes' to PLOS Genetics.

The manuscript was fully evaluated at the editorial level and by independent peer reviewers. The reviewers appreciated the attention to an important topic but identified some concerns that we ask you address in a revised manuscript

We therefore ask you to modify the manuscript according to the review recommendations. Your revisions should address the specific points made by each reviewer.

[LINK]

Yours sincerely,

Alex Buerkle

Associate Editor

PLOS Genetics

Kirsten Bomblies

Section Editor: Evolution

PLOS Genetics

This manuscript has been reviewed thoroughly by three referees, all of whom find the work interesting and compelling. The reviews offer suggestions for improvement of the manuscript, including some requests for possible additions to the analyses, which I encourage the authors to consider. Reporting a false positive rate along with the power analysis seems like a good idea. The reviews are also very clear in their praise for the clarity of the presentation and the important contribution this work will make to how we think about and analyze trait covariation across species. I agree and think this manuscript will be read with lots of interest.

As a minor point, I will add that it would be good to emphasize the time scale a bit more in the introduction. For example, on line 92, perhaps "To address the effects of introgression on quantitative traits" could be modified to "To address the effects of historical introgression on quantitative traits". I think this is worthwhile because people study trait variation due to contemporary admixture and introgression regularly with admixture mapping. The text is clear that this manuscript is about phylogenomics, deeper timescales reflected on trees, and comparative methods, but I believe readers will be benefit from a very explicit statement in the introduction of the relevant time scale.

Reviewer's Responses to Questions

**Comments to the Authors:**

Reviewer #1: The paper by Hibbins and Hahn proposes a statistical framework for estimating the amount of introgression between two species while accounting for incomplete lineage sorting (ILS). Both ILS and introgression are common in evolution and failure to account for them can influence inferences of the rate, degree of convergence, or amount of selection that has occurred on a particular trait. As such, this work is an important step towards a more complete understanding of the evolution of quantitative traits. The paper is clearly written and the results presented in an easy to follow manner. However, I do have a couple of concerns about the current presentation.

1. The authors did not present data on false positive rates and this information is crucial for understanding the power analyses performed. This data should already be included in the simulations, so additional simulations aren’t needed.

2. The power analysis comparing a model of ILS only and of ILS + introgression used a single set of parameters and the values chosen for some parameters (e.g. number of genes and introgression amount) are more conducive to identifying a signal of introgression than what is presented in the power analysis of just the ILS+introgression model or the empirical data analyzed. It would be useful to see the results of this comparison for a range of conditions, especially those that covered what would be expected for the empirical data to better understand the ability to distinguish between these two models.

3. It is hard to tell exactly, but the power analyses performed suggest relatively low power to detect introgression in the empirically analyzed data using the ILS+introgression approach. This is particularly true for the ‘low’ data set, yet introgression is found. A discussion or resolution of this discrepancy would be useful to readers.

4. The presented Q3 statistic allows one to estimate the extent and species involved in introgression. However, as currently presented it is not integrated into a framework that allows the estimation of the rate of trait evolution or the degree of ILS. Maybe I’m mistaken, but the models seem to be nested, with the ILS only model being a special case of the ILS+introgression model. This would suggest that a likelihood ratio approach could be used to determine whether the inclusion of introgression in the model is actually warranted by the data. I recognize that this was not the approach taken by the authors, but this seems like a limitation to the current approach that would be useful to mention.

Minor point

Line 482 – missing location of script

Reviewer #2: This paper develops an approach to examine gene expression variation and covariation between species that takes into account both incomplete lineage sorting and introgression. The paper then goes to show that gene expression covariation between species that have been subject to introgression also show the effect in gene expression. They show this in particular within 2 subclades in Solanum that show varying levels of introgression in the genome.

The modelling seems sound although I do not have the expertise to comment on this deeply. However, I do think this is a very interesting area of study as we begin to grasp the multilayered effects of interspecies hybridization on evolution. The results with gene expression data do show a significant effect of introgression, although the magnitude of the difference was not great (476 vs. 449 genes).

My one concern is that the authors really only apply their method to 2 groups – the low and high triplet clades they identify. It would have been much stronger if they could find other comparisons, as there will always be a question of whether this particular high triplet is a fluke. Conversely, maybe showing what the results look like in clades that have no evidence of introgression may also provide greater confidence.

Reviewer #3: The evolution of gene expression has been an important source of research in the last three decades. Nonetheless, most, if not all studies of gene expression, ignore the effect that introgression might have on the distribution of gene expression as a trait in phylogenetic trees. The first part of the paper addresses this very gap and develops an analytical framework to recapitulate the evolution of quantitative traits following a Brownian motion model of evolution in the face of introgression and incomplete lineage sorting. The piece provides a derivation of the expected trait variances and covariances among hybridizing species using the multispecies network coalescent as a framework. The methods are clear, the appendix is welcome and the results are well presented. The authors are forward and state that while are other models of trait evolution but elaborate on the reasons to not expand their model. The model is elegant and timely and I think will be an excellent addition to the literature. The authors present a fairly large simulation analyses in Figure 3 which also bolster the importance of the piece.

The second part of the paper takes the theoretical prediction and uses the model to evaluate the evolution of ovule gene expression 12 species of Solanum. Using the model described above, the authors find that gene expression similarity is correlated with the tree topology of protein-coding genes which is consistent with cis-acting effects of introgressed variation on expression. To my knowledge, this is one of the finest assessments of the evolution of gene expression in diverging species because instead of focusing only on reporting the results, there is an underlying model and a potential explanation of the reasons that lead to such patterns.

The manuscript is exceptionally well written, has an excellent level of scholarship, is well contextualized, has clear results, and the discussion points out the relevant caveats. Overall, the piece is a timely and excellent contribution to the literature and I am certain will have an important effect on how we understand the evolution of gene expression, speciation, and introgression altogether.

I just have a few comments, all of which are rather minor and are meant to help the readability of the piece.

The colors in Figure 1 are hard to follow, especially for a color-blind person.

Line 282. I think “thousands of quantitative traits’ refers to the expression of individual genes. I would rephrase this sentence to be more clear.

Line 499. Elaborate on the rationale for this ordination.

**Have all data underlying the figures and results presented in the manuscript been provided?**

Reviewer #1: Yes

Reviewer #2: Yes

Reviewer #3: None

PLOS authors have the option to publish the peer review history of their article (what does this mean?). If published, this will include your full peer review and any attached files.

Reviewer #1: No

Reviewer #2: No

Reviewer #3: No

---

## [Editor Report · Decision Letter 1]

18 Oct 2021

Dear Dr Hibbins,

We are pleased to inform you that your manuscript entitled "The effects of introgression across thousands of quantitative traits revealed by gene expression in wild tomatoes" has been editorially accepted for publication in PLOS Genetics. Congratulations!

Yours sincerely,

Alex Buerkle

Associate Editor

PLOS Genetics

Kirsten Bomblies

Section Editor: Evolution

PLOS Genetics

Comments from the reviewers (if applicable):

I appreciate the authors' clear and comprehensive responses to the suggestions and comments from the previous round of review. The manuscript is likely to prompt further empirical and methodological advances and represents a clear contribution to the field.

**Data Deposition**

http://datadryad.org/submit?journalID=pgenetics&manu=PGENETICS-D-21-01009R1

**Press Queries**

---

## [Editor Report · Acceptance letter]

27 Oct 2021

PGENETICS-D-21-01009R1 

The effects of introgression across thousands of quantitative traits revealed by gene expression in wild tomatoes 

Dear Dr Hibbins, 

We are pleased to inform you that your manuscript entitled "The effects of introgression across thousands of quantitative traits revealed by gene expression in wild tomatoes" has been formally accepted for publication in PLOS Genetics! Your manuscript is now with our production department and you will be notified of the publication date in due course.

With kind regards,

Agnes Pap

PLOS Genetics

On behalf of:
